# Go beyond End-to-End Training: Boosting Greedy Local Learning with Context Supply

## Abstract

Traditional end-to-end (E2E) training of deep networks necessitates storing intermediate activations for back-propagation, resulting in a large memory footprint on GPUs and restricted model parallelization. As an alternative, greedy local learning partitions the network into gradient-isolated modules and trains supervisely based on local preliminary losses, thereby providing asynchronous and parallel training methods that substantially reduce memory cost. However, empirical experiments reveal that as the number of segmentations of the gradient-isolated module increases, the performance of the local learning scheme degrades substantially, severely limiting its expansibility. To avoid this issue, we theoretically analyze the greedy local learning from the standpoint of information theory and propose a ContSup scheme, which incorporates context supply between isolated modules to compensate for information loss. Experiments on benchmark datasets (i.e. CIFAR, SVHN, STL-10) achieve SOTA results and indicate that our proposed method can significantly improve the performance of greedy local learning with minimal memory and computational overhead, allowing for the boost of the number of isolated modules.

## 1 Introduction

End-to-end (E2E) back-propagation, a standard training paradigm for deep neural networks, enables deep neural networks to solve complex tasks and cognitive applications with great success (Szegedy et al., 2015; He et al., 2016; Huang et al., 2016). As shown in Figure 1a, an E2E training loss is calculated at the final layer, and the error is propagated backward layer-by-layer for weights update. In this case, the E2E is caught in the well-known backward-locking problem (Jaderberg et al., 2017; Frenkel et al., 2021; Duan & Principe, 2022), which prohibits module updates until all dependent modules have completed forward and backward passes, and restricts the network from performing training in a sequential manner (Jaderberg et al., 2017). During the forward pass, intermediate tensors and operations required for weights update must be preserved, resulting in a high memory cost and frequent memory access (Mostafa et al., 2018). In general, memory constraints impede the training of state-of-the-art DNNs with high-resolution inputs and large batch sizes, and the strong inter-layer backward dependency prevents training parallelization, thereby delaying the training process (Chen et al., 2016; Gomez et al., 2017).

As an alternative to E2E training, numerous local learning paradigms have been proposed to optimize gradient computations and weight updates by cutting off the feedback path for greater memory efficiency and model parallelization (Hinton et al., 2006; Bengio et al., 2006; Akrout et al., 2019; Meulemans et al., 2020; Belilovsky et al., 2020; Duan & Principe, 2022). A representative example is the greedy local learning (GLL) method (Löwe et al., 2019), which partitions a deep neural network into multiple gradient-isolated modules and trains them independently under local supervision (see Figure 1b). Since back-propagation occurs only within local modules, it is unnecessary to store all intermediate activations simultaneously. In addition, it is feasible to train local modules in parallel since error signals from subsequent layers are no longer required (Chen et al., 2016; Mostafa et al., 2018; Huo et al., 2018; Belilovsky et al., 2020). This approach is also considered more biologically plausible due to the fact that biological systems are highly modular and primarily learn from local signals (Crick, 1989; Dan & Poo, 2004; Bengio et al., 2016); and parallel pathways frequently perform distinct but somewhat overlapping computations(Patel et al., 2023). However, in contrast to E2E training, GLL suffers from a performance drop issue (Mostafa et al., 2018; Belilovsky et al.,

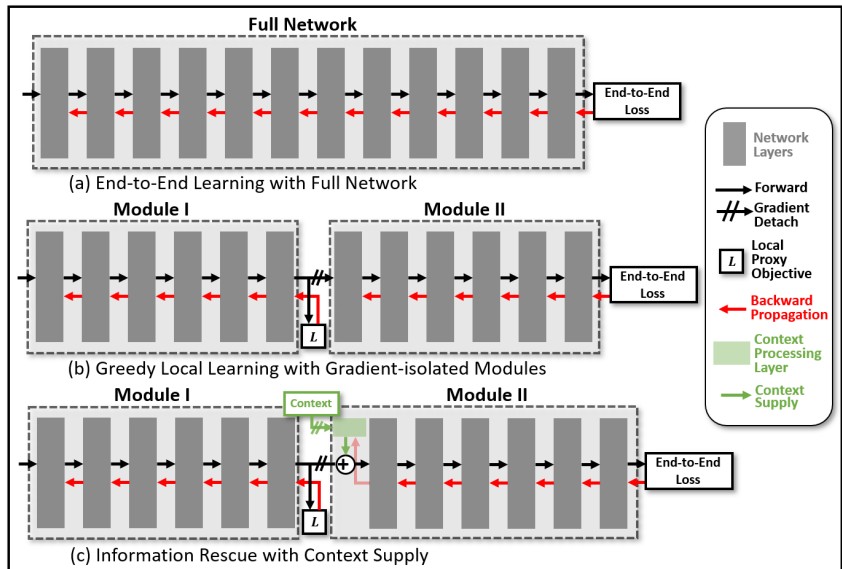

Figure 1: **Dataflow in training schemes.** (a) The standard paradigm that back-propagates errors end-to-end in reverse. (b) Greedy learning occurred with local-defined objectives. (c) ContSup provides the context path in addition to feature paths, and consists of two portions with element-wise addition to preserve the same shapes of features and computations within modules.

2019; Wang et al., 2021; Duan & Principe, 2022), and an increase in the number of segments correlates directly with a large performance decline (Wang et al., 2021; Guo et al., 2023).

Theoretical support for the design and optimization of GLL can be derived from an information-theory-based analysis (Ma et al., 2019; Wang et al., 2021; Du et al., 2021). It is discovered through empirical experiments that local modules tend to generate more discriminative intermediate features at earlier network layers, resulting in task-relevant information decreasing with network depth (Wang et al., 2021; Guo et al., 2023). In order words, the short-sighted GLL tends to learn intermediate features that only benefit prior local objectives, disregarding the requirements of the remaining layers (Wang et al., 2021; Du et al., 2021). By introducing the additional objective of maximizing the entropy of intermediate features, the short-sight problem can be improved (Wang et al., 2021; Zhang et al., 2022; Ma et al., 2019); however, this method of constructing constraints frequently requires good priors and careful adjustments, which is highly constructive. Moreover, since the phenomenon of information loss is irreversible, the upper bound of the potency of the intermediate features decreases with depth, which limits the behaviors of the deep modules to the incomplete supply of the early modules and cannot be recovered by well-designed local objectives.

Based on the above observations, we proceed with the theoretical analysis of GLL and postulate that irreversible information loss is the crucial bottleneck restricting the overall network performance. Then, we intuitively propose the Context Supply (ContSup) scheme (Figure 1c) to supplement the context for the intermediate feature allowing it to access a portion of the lost information and subsequently escape the dilemma. Empirical experiments show that the proposed ContSup can effectively compensate for the information loss of intermediate features while maintaining a high level of performance with a large number of isolated modules, and yields state-of-the-art GLL results on three benchmarks (i.e. CIFAR-10(Krizhevsky, 2009), SVHN(Netzer et al., 2011) and STL-10(Coates et al., 2011)).

## 2 THEORETICAL ANALYSIS OF GREEDY LOCAL LEARNING

In this section, we take insights from information theory to present a theoretical framework for greedy local learning, showing that the irreversible loss of task-relevant mutual information is a bottleneck of the final decision. We also analyze the benefits and drawbacks of the current mainstream ideas for optimizing GLL based on defining a local reconstruction objective and lay the theoretical foundation for our further optimization.

## 2.1 PRELIMINARY

The greedy local learning (GLL) method is proposed to divide a deep network into several gradient-isolated modules and trains them separately under local objectives (Algorithm 1). A series of $\mathcal{F}^l$ modules with main parameters $\theta_l$ forms a feedforward network with $L$-partitioned gradient-isolated modules, in which the network's primary parameters $w_l$ are trained locally alongside a local auxiliary module $\mathcal{A}^l$. The predictive feature, $\hat{y}_l$, is obtained by the local auxiliary module to evaluate the performance of the local classification task with the objective function $\hat{\mathcal{L}}$. Consistently, we denote the network's final result as $\hat{y}_L$, which is the output of the final $L^{th}$ module via its implicit auxiliary module $\mathcal{A}^L$ (which is considered within the entire network). The final objective $\hat{\mathcal{L}}(\hat{y}_L, y)$ is then delivered in the same manner as E2E training.

---

**Algorithm 1:** Greedy Local Learning.

**Input:** Number of gradient-isolated modules $L$; Datasets $\mathcal{D}$; Epochs $T$; Batch size $B$.

1 **Initialize** Paramters $\{\theta_l, w_l\}_{l \leq L}$ ;
2 **for** $t = 1$ *to* $T$ **do**
3     Sample a mini-batch $\{h_0^b, y^b\}_{b \leq B}$ over $\mathcal{D}$ ;
4     **for** $l = 1$ *to* $L$ **do**
5        $h_l^b \leftarrow \mathcal{F}_{\theta_l}^l(h_{l-1}^b)$ ;
6        $\hat{y}_l^b \leftarrow \mathcal{A}_{w_l}^l(h_l^b)$ ;
7        $(\theta_l, w_l) \leftarrow$ Update by $\nabla_{\theta_l, w_l} \hat{\mathcal{L}}(\hat{y}_l^b, y^b; \theta_l, w_l)$ ;
8     **end**
9 **end**

---

## 2.2 NAÏVE GREEDY LOCAL LEARNING SUFFERS A DILEMMA CALLED CONFIRMED HABITS

We first clarify that the performance discrepancy between GLL and E2E schemes is the result of a confirmed habit dilemma in feature transfer, which we attribute to an information bottleneck in GLL's working mechanism.

**Notation of mutual information about features.** We focus on the information captured by the intermediate feature $h_l$, and let $I(h_l, y)$ denote the amount of task-relevant information in $h_l$. Meanwhile, the task-irrelevant information in the input data $x$ can be formed by introducing the concept of nuisance $r$ (Achille & Soatto, 2018), such that the Markov chain stands as $(y, r) \rightarrow x \rightarrow h$ (more details in Appendix A).

**The trend of information loss is conclusive and irreversible.** From the perspective of information theory, it is clear to deduce that the serial inference of isolated modules gradually reduces the information entropy, and the amount of task-related information it contains will inevitably diminish (decreasing potency); moreover, since the local objective of auxiliary modules is defined in terms of the prediction target $\hat{y}_l$, it cannot act directly on the mutual information $I(h_l, y)$ of the intermediate features, resulting in the loss of task-related information (local short-sight); we restate those two conclusions in Theorem 1 (full proof in Appendix B).

**Theorem 1.** *The feedforward process of isolated modules forms the Markov chain* $(y, r) \rightarrow x \rightarrow \dots \rightarrow h_{l-1} \rightarrow h_l \rightarrow \hat{y}_l$ *in Greedy Local Learning. Then the decreasing trend of mutual information is given by:*

$$I(x, y) \geq I(h_{l-1}, y) \geq I(h_l, y) \geq I(\hat{y}_l, y) \tag{1}$$

*which can be further divided into two clear insights:*
*1.1 Decreasing potency. The upper bound of task-related information is decreasing over isolated modules:*

$$I(x, y) \geq I(h_{l-1}, y) \geq I(h_l, y) \tag{2}$$

*1.2 Local short-sight. The local prediction always lags the task-related information of features:*

$$I(h_l, y) \geq I(\hat{y}_l, y) \tag{3}$$

Experiments indicate that $I(h_l, y)$ will decrease substantially during the greedy module process, whereas it will remain essentially unchanged during the E2E process (Wang et al., 2021; Du et al., 2021). Therefore, the loss of task-related information will accumulate in the cascade (Equation 2), resulting in a gradual decline in the final performance potential of the network. The final prediction result $I(\hat{y}_L, y)$ is fundamentally constrained by the sustained irreversible loss on $I(h_l, y)$, falling into the confirmed habit dilemma (see Figure 2a).

**Local learning depends on the progressive relay.** From a different angle, GLL can be viewed as the integrated learning of a series of classifiers, making the incremental improvement in its perfor-

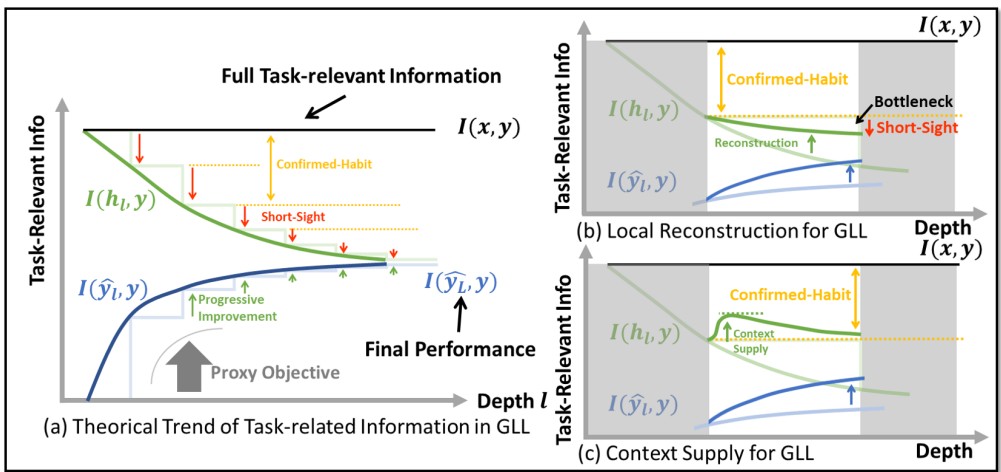

Figure 2: **An information-theoretic perspective in GLL.** (a) illustrates information trends via network depth, showing the monotonically decreasing of $I(h_l, y)$ and general upward trend of $I(\hat{y}_l, y)$, where the final performance is obtained by progressive improvement. (b) shows that local reconstruction efforts to alleviate the short-sight issue are impeded by the confirmed habit. (c) illustrates the function of context that enables a local module to surpass its obstruction.

mance more intuitive. Under mild conditions, the series of classifiers improve the training error at each module as shown below (full proof in Appendix B):

**Theorem 2 (Progressive improvement).** *Assume that $\mathcal{F}_{\theta_l}^l$ can be potentially equivalent as an identity mapping as $\mathcal{F}_{\theta_l}^l(h_{l-1}) = h_{l-1}$, then there exists $\hat{\theta}_l$ such that:*

$$\hat{\mathcal{L}}_l\left(\hat{y}_l, y; \theta_l, w_l\right) \leq \hat{\mathcal{L}}_l\left(\hat{y}_l, y; \hat{\theta}_l, w_{l-1}\right) = \hat{\mathcal{L}}_{l-1}\left(\hat{y}_{l-1}, y; \theta_{l-1}, w_{l-1}\right) \tag{4}$$

*Once the local loss $\hat{\mathcal{L}}_l$ is defined on task-relevant mutual information, we obtain:*

$$I\left(\hat{y}_{l-1}, y\right) \leq I\left(\hat{y}_l, y\right) \tag{5}$$

In practice, a technical requirement for the actual optimization procedure is not to generate an objective that is inferior to the initialization, which can be met by selecting the optimal solution along the optimization trajectory. Stacking solitary models can therefore facilitate the output of the auxiliary module and incrementally enhance the mutual information of $y_l$ (Equation 5).

**Dilemma of confirmed habits.** Based on the preceding discussion, we can determine the trend of task-related information in GLL (Figure 2a), and incorporate the presented propositions together:

**Theorem 3.** *Combined with Equation 1-5, the final performance of the overall network relies on task-relevant information $I(\hat{y}_L, y)$, which is:*

- *Developed by accumulating efforts from series local learning:*

$$I(\hat{y}_{l-1}, y) \leq I(\hat{y}_l, y) \leq ... \leq I(\hat{y}_L, y) \tag{6}$$

- *Bounded by continuous hurt from series local learning [confirmed habits]:*

$$I(x, y) \geq I(h_{l-1}, y) \geq I(h_l, y) \geq I(h_L, y) \geq I(\hat{y}_L, y) \tag{7}$$

Theorem 3 shows a dilemma of GLL that when the network depth increases, even if we can find a careful step for progressive improvement (Equation 6), our potency (theoretical upper bound of performance) is irreparably damaged, trapping us in the dilemma of confirmed habits (Equation 7). That is, the more gradient-isolated modules a network is divided into, the more it will be affected by the confirmed habit. Therefore, in order to obtain a high level of efficacy, the experiment must ensure that $L$ is as small as possible, which severely restricts the scalability of the GLL method.

## 2.3 THE LOCAL RECONSTRUCTION EASES SHORT-SIGHT BUT STILL IN CONFIRMED HABITS.

Several works propose assuring the original input's information ($I(h, x)$) as part of the learning objective to prevent information loss caused by excessive local optimization (Ma et al., 2019; Wang et al., 2021; Zhang et al., 2022), which is typically implemented by reconstructing from features

to input. Since $I(h, x)$ cannot be directly calculated, pursuing the local maximization of $I(h, x)$ is equivalent to the reconstruction task for the original image, i.e., by simultaneously training an auxiliary decoder from $h$ to $x$. Those reconstruction methods have been shown to be effective by empirical experiments (Wang et al., 2021) and are regarded as a solution to the local short-sight problem (Figure 2b); therefore, analyzing their working mechanism can assist us in better dissecting and comprehending the GLL method. A plausible explanation is that reconstruction methods build auto-encoders to facilitate the extraction of essential features. Notably, beginning with information theory, we can determine that during deterministic feature transmission, $I(h, x)$ is equivalent to information entropy $H(h)$. The optimization of $I(h, x)$ is therefore identical to maximizing the decreasing upper bound of local information entropy:

**Lemma 4 (Entropy Bound).** *Since the Markov chain $x \rightarrow ... \rightarrow h_{l-1} \rightarrow h_l$ is the deterministic procedure in the network forward, the input-related information is bounded by:*

$$H(x) \geq H(h_{l-1}) = I(h_{l-1}, x) \geq H(h_l) = I(h_l, x) = I(h_l, h_{l-1}) \tag{8}$$

Lemma 4 shows that increasing $I(h_l, x)$ by defining the reconstruction target of $h_l$ is identical to increasing the information entropy $H(h_l)$, whose upper bound is given by the information entropy $H(h_{l-1})$ of the previous module $h_{l-1}$. Therefore, constructing a decoder from $h_l$ to $h_{l-1}$ is sufficient to satisfy the requirement of increasing information entropy in practice, and theoretically works the same as a decoder for origin input $x$, as confirmed in a comparable scheme for performance and memory optimization (Zhang et al., 2022).

Clearly, methods of maximizing information entropy can reduce information loss, which simultaneously reduces loss of nuisance $I(h, r)$ and task-relevant information $I(h, y)$ (details in Appendix B); thus, we conclude their success in optimizing short-sight and indicate they are still constrained by the upper bound $I(h_l, y) \leq I(h_{l-1}, y)$ (Equation 7), resulting in a decline in overall performance when the number of isolated-partitions is high and being trapped in confirmed habits that restricted by previously lost information.

## 3 CONTEXT SUPPLY GIVES CHANCE TO ESCAPE THE DILEMMA

When a new gradient-isolated module needs to be layered on top of the existing network structure and task-related information in superficial layers has been lost, the confirmed habit dilemma cannot be avoided, i.e., with the irreversible loss in the intermediate feature $h_l$, the information supremum of extracted $h_{l+1}$ has been tightly framed:

$$I(\hat{y}_{l+1}, y) \leq sup\{I(\hat{y}_{l+1}, y)\} = I(h_{l+1}, y) \leq sup\{I(h_{l+1}, y)\} = I(h_l, y) \tag{9}$$

If we continue to increase the number of greedy modules on top of this, the maximum theoretical result cannot surpass $I(h_l, y)$, and the loss of information may even be worsened by the short-sighted objective. In the case of a large number of network segments, the serial structure of GLL imposes significant limitations, as the task-relevant information is directly reduced, preventing cascading on a large scale and limiting the method's generality and scalability. On this basis, we propose the ContSup structure to incorporate an additional information path in an attempt to supplement the lost information by context, aimed to escape the discussed dilemma (Figure 2c).

**Context supply to intermediate features.** Assuming we have context $c_l$ with $I(c_l, y) \geq I(h_{l-1}, y)$, to compensate for the lost task information, we hope to obtain integrated feature $h_{l-1}^c = h_{l-1} + \mathcal{M}^l(c_l)$, such that there exists proper $\mathcal{F}^l$ and $\mathcal{M}^l$ for:

---

**Algorithm 2:** GLL with ContSup.

1 **Initialize** Parameters $\{\theta_l, w_l, \phi_l\}_{l \leq L}$;

2 **for** $t = 1$ *to* $T$ **do**

3     Sample a mini-batch $\{h_0, y\}_B$ over $\mathcal{D}$;

4     **for** $l = 1$ *to* $L$ **do**

5         $c_l \leftarrow$ context selection;

6         $h_{l-1}^c \leftarrow h_{l-1} + \mathcal{M}_{\phi_l}^l(c_l)$;

7         $h_l \leftarrow \mathcal{F}_{\theta_l}^l(h_{l-1}^c)$;

8         $\hat{y}_l \leftarrow \mathcal{A}_{w_l}^l(h_l)$;

9         $(\theta_l, w_l, \phi_l) \leftarrow$ Update by $\nabla_{\theta_l, w_l} \hat{\mathcal{L}}(\hat{y}_l, y; \theta_l, w_l, \phi_l)$;

10     **end**

11 **end**

---

$$sup\{I(h_l, y)\} = I(h_{l-1}^c, y) \geq I(h_{l-1}, y) \tag{10}$$

following the trend in Figure 2c. For simplicity, we utilize element-wise addition to incorporate context and feature without changing the shape size of $h_l$ and the configuration of $\mathcal{F}^l$, i.e.,

$h_l = \mathcal{F}^l(h_{l-1}^c) = \mathcal{F}^l(h_{l-1} + \mathcal{M}^l(c_l))$, so the context-supply under this definition can be readily transferred to the existing GLL framework.

**Concise priors to ascend supremum.** There are currently two simple and intuitive options available for locating a suitable context to boost $l^{th}$ module after $(l-1)^{th}$ module:

**1. Introduce the origin input $x$ directly to supplement task-relevant information.** Let $c_l = x$, and

$$h_l = \mathcal{F}^l(h_{l-1}^c) = \mathcal{F}^l\left(h_{l-1} + \mathcal{M}_E^l(x)\right) \tag{11}$$

where $\mathcal{M}_E^l$ works as a local encoder to compress input $x$ into the same size as $h_l$. Then, its supreme is extended into:

$$I(h_l, y) \leq sup\{I(h_l, y)\} = I(h_{l-1}^c, y) \leq I(x, y) \tag{12}$$

**2. Introduce a shortcut connection between the adjacent hidden feature,** i.e., let $c_l = h_{l-2}^c$ to supplement the task-relevant information that may be lost when $h_{l-2}^c \rightarrow h_{l-1}$, and

$$h_l = \mathcal{F}^l(h_{l-1}^c) = \mathcal{F}^l(h_{l-1} + \mathcal{M}_{R_1}^l(h_{l-2}^c)) \tag{13}$$

which is similar to shortcut connections across modules, where $\mathcal{M}_{R_1}^l$ is used to align feature shapes. In this case, the supreme is then:

$$I(h_l, y) \leq sup\{I(h_l, y)\} = I\left(h_{l-1}^c, y\right) \leq I\left(h_{l-2}^c, y\right) \tag{14}$$

Clearly, both context designs go beyond the cascaded Markov-process $h_{l-2} \rightarrow h_{l-1} \rightarrow h_l$; consequently, the assumption of theorems (Equation 1) is refuted in this instance, so that $I(h_l, y) \geq I(h_{l-1}, y)$ may hold in some cases, allowing the opportunity to flee the confirmed habits dilemma.

Empirical experiments show that the above two priors can simply and effectively protect against the loss of $I(h_l, y)$ during transmission, and enhance the performance of GLL by incurring a small additional calculation and memory overhead. Simultaneously, these two methods can effectively mitigate the trend of performance degradation as the number of divided modules increases, thereby reducing memory overhead substantially while maintaining high performance.

Table 1: Comparison of GLL with different context modes. The experiment is evaluated on CIFAR-10 with 16-partitioned ResNet-32 (see Appendix C for details). "base" refers to GLL without context.

|  | base | E | R1 | R1E |
|---|---|---|---|---|
| Test Error | 16.21% | 10.32% | 15.30% | 10.14% |

**Topological extension from priors.** On the basis of efficacy, we can intuitively combine the two methods, that is, introduce $\mathcal{M}_E^l(x)$ and $\mathcal{M}_{R_1}^l(h_{l-2}^c)$ simultaneously to construct for better performance in practice:

$$h_{l-1}^c = h_{l-1} + \mathcal{M}_E^l(x) + \mathcal{M}_{R_1}^l(h_{l-2}^c) \tag{15}$$

To avoid losing scalability, we consider whether it is possible to achieve further accuracy improvement by constructing a path with all previous information, i.e., by constructing the largest topological connection across all modules on the original structure to construct:

$$h_l^c = h_l + \mathcal{M}_R^{l+1}\left([h_{l-1}^c, \ldots, h_1]\right) + \mathcal{M}_E^{l+1}(x) = h_l + \mathcal{M}_E^{l+1}(x) + \sum_i \mathcal{M}_{R_i}^{l+1}\left(h_{l-i}^c\right) \tag{16}$$

In this way, we can be very confident that $h_l^c$ contains all the required information, despite the fact that it introduces a substantial amount of calculation and memory overhead.

The emphasis of the experimental portion of this article will be on the two simplest priori assumptions, denoted as [E] (Equation 11) and [R1] (Equation 13) respectively, and their combination as [R1E] (Equation 15) to prove that the ContSup structure is simple but effective; additionally, we will prove the feasibility of introducing topological connections through experiments ([RnE], Equation 16), and demonstrate that the proposed structure can be further flexibly extended.

## 4 EXPERIMENTS

In this section, we first demonstrate the effectiveness of the proposed ContSup in improving accuracy and optimizing GPU memory cost via comparative experiments on benchmarks. Then, we combine the ablation experiments to investigate the effect of structure and hyperparameter settings in ContSup. Finally, we will discuss the scalability of ContSup and analyze the benefits and tradeoffs of the potential expansion mode.

Table 2: Performance of GLL methods on CIFAR-10 with $K$-partitioned ResNet-32. The averaged test errors and standard deviations of 5 independent trials on ContSup are reported. The best result of each case is shown in bold.

| $K$ | DGL | InfoPro (softmax) | InfoPro (contrast) | BackLink (best result) | ContSup[E] (softmax) | ContSup[R1E] (softmax) | ContSup[E] (contrast) | ContSup[R1E] (contrast) |
|---|---|---|---|---|---|---|---|---|
| 2 | $8.69 \pm 0.12\%$ | $8.13 \pm 0.23\%$ | $7.76 \pm 0.12\%$ | **6.97%** | $8.04 \pm 0.22\%$ | $7.97 \pm 0.28\%$ | $7.87 \pm 0.24\%$ | $7.85 \pm 0.16\%$ |
| 4 | $11.48 \pm 0.20\%$ | $8.64 \pm 0.25\%$ | $8.58 \pm 0.17\%$ | **7.55%** | $8.75 \pm 0.29\%$ | $8.42 \pm 0.12\%$ | $8.52 \pm 0.21\%$ | $8.09 \pm 0.15\%$ |
| 8 | $14.17 \pm 0.28\%$ | $11.40 \pm 0.18\%$ | $11.13 \pm 0.19\%$ | 10.6% | $9.67 \pm 0.26\%$ | $9.39 \pm 0.24\%$ | $9.57 \pm 0.18\%$ | **$9.32 \pm 0.19\%$** |
| 16 | $16.21 \pm 0.36\%$ | $14.23 \pm 0.42\%$ | $12.75 \pm 0.11\%$ | 12.4% | $10.52 \pm 0.43\%$ | $10.09 \pm 0.26\%$ | $10.21 \pm 0.45\%$ | **$9.98 \pm 0.18\%$** |

## 4.1 SETUP

**Environments and Evaluation.** We evaluate the efficiency of proposed ContSup on three image datasets (CIFAR-10 (Krizhevsky, 2009), SVHN (Netzer et al., 2011), and STL-10 (Coates et al., 2011)) with ResNet (He et al., 2016) as the foundational network architecture. To train networks with GLL, the entire network is divided into $K$ gradient-isolated modules containing the same number of layers. The interior modules are trained with locally designed objectives, whereas the final module is trained directly with the standard E2E loss. Details of data processing, network setting, and training configurations are contained in Appendix C.

**Implementation modes of context supply** are notated in the form 'ContSup[RnE]', which indicates how context is selected, where (1) if the original image is included in context, then denote with 'E' and (2) the $n$ number of adjacent used in, as 'R$n$'. The optimal representations of two simple basics are 'E' as the input-encoded context and 'R1' as the only context of the last feature. Notably, decoupled greedy learning (DGL) (Belilovsky et al., 2019) is the simplest implementation of GLL and can be viewed as the baseline case 'R0' of ContSup, in which no context is provided.

## 4.2 COMPARISON WITH STATE-OF-THE-ARTS

**Quick Comparisons.** We first compare ContSup with five recently proposed algorithms: decoupled greedy learning (DGL) (Belilovsky et al., 2020), BoostResNet (Huang et al., 2018), deep incremental boosting (DIB) (Mosca & Magoulas, 2017), InfoProp (Wang et al., 2021), and BackLink (Guo et al., 2023) in Figure 3. Notably, DGL is the simplest implementation of GLL and can be regarded as the baseline method for InfoPro, BackLink, and our ContSup. As previously indicated, we make direct changes to the proposed ContSup on DGL by including a simple but functional context route. For the sake of brevity, we only present the best results reported in the corresponding papers for a quick comparison, despite the fact that each approach may have detailed structures and hyperparameters. One can observe that, when $K$ increases, the performance of several different approaches shows a clear negative trend, indicating that the number of segmentations poses an essential obstacle to the further expansion of GLL

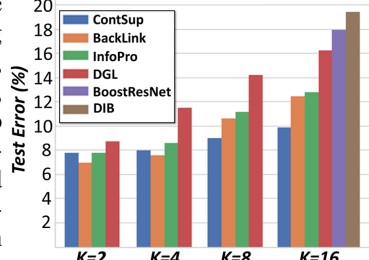

Figure 3: Comparisons of ContSup and state-of-the-art GLL methods in terms of the test errors. The best results of methods based on $K$-partitioned ResNet-32 and CIFAR-10 are reported.

methods; in this case, the performance of ContSup remains reasonably high, indicating the value of context supply in resolving the GLL bottleneck problem.

**Comprehensive results with benchmarks** are shown in Table 2 and Table 3. We compare ContSup's performance to that of DGL, InfoPro, and BackLink on a variety of image classification benchmarks. Specifically, the GLL scheme requires a local classifier that is typically defined by the objective function of cross-entropy based on softmax, while InfoPro (Wang et al., 2021) suggested using contrastive-like loss (Chen et al., 2020; He et al., 2020) as the local objective in GLL to improve performance; consequently, we set ContSup to use both cases (designated in softmax/contrast) as controls. On the premise of the baseline method (DGL), it is discovered that ContSup can further enhance performance, especially when $K$ is large. We found, however, that when $K$ is small (i.e., $K = 2/4$), the performance gain brought by ContSup is not readily apparent; we consider that the impact caused by the confirmed habit dilemma to be negligible at that time due to the small number of isolated nodes, so the mutual information gain brought by context has no significant effect. (Detailed analysis in Appendix C).

**GPUs memory Efficiency.** Figure 4 compares the GPU memory footprint of ContSup to those of InfoPro, DGL, and E2E, and illustrates the linkages of error rates trade-offs as functions by joining

Table 3: Comparison of GLL methods on benchmarks with $K$-partitioned ResNet-110. The averaged test errors and standard deviations of 5 independent trials on ContSup are reported. The best result of each case is shown in bold.

| Datasets | $K$ | DGL | InfoPro (softmax) | InfoPro (contrast) | BackLink (best result) | ContSup[R1E] (softmax) | ContSup[R1E] (contrast) |
|---|---|---|---|---|---|---|---|
| CIFAR-10 (E2E:6.50 ± 0.34%) | 2 | 7.70 ± 0.28% | 7.01 ± 0.34% | 6.42 ± 0.08% | **6.36%** | 7.40 ± 0.22% | 6.56 ± 0.10% |
| | 4 | 10.50 ± 0.11% | 7.96 ± 0.06% | 7.30 ± 0.14% | 7.79% | 7.81 ± 0.20% | **7.18 ± 0.12%** |
| | 8 | 12.46 ± 0.37% | 9.40 ± 0.27% | 8.93 ± 0.40% | 9.25% | 8.46 ± 0.32% | **7.66 ± 0.30%** |
| | 16 | 13.80 ± 0.15% | 10.78 ± 0.28% | 9.90 ± 0.19% | 9.75% | 8.91 ± 0.32% | **8.89 ± 0.24%** |
| SVHN (E2E:3.07 ± 0.23%) | 2 | 3.61 ± 0.16% | 3.41 ± 0.08% | **3.15 ± 0.03%** | 3.35% | 3.21 ± 0.10% | 3.23 ± 0.03% |
| | 4 | 4.97 ± 0.19% | 3.72 ± 0.03% | **3.28 ± 0.11%** | 4.33% | 3.54 ± 0.03% | 3.41 ± 0.08% |
| | 8 | 5.35 ± 0.13% | 4.67 ± 0.07% | 3.62 ± 0.11% | 4.67% | 3.98 ± 0.10% | **3.45 ± 0.08%** |
| | 16 | 5.55 ± 0.34% | 5.14 ± 0.08% | 3.91 ± 0.16% | 4.91% | 4.03 ± 0.22% | **3.91 ± 0.23%** |
| STL-10 (E2E:22.27 ± 1.61%) | 2 | 24.96 ± 1.18% | 21.02 ± 0.51% | 20.99 ± 0.64% | **20.40%** | 20.87 ± 0.66% | 21.20 ± 0.78% |
| | 4 | 26.77 ± 0.64% | 21.28 ± 0.27% | 22.73 ± 0.40% | 23.72% | 21.96 ± 0.28% | **20.53 ± 0.03%** |
| | 8 | 27.33 ± 0.24% | 23.60 ± 0.49% | 25.15 ± 0.52% | 24.16% | 23.61 ± 0.04% | **23.38 ± 0.43%** |
| | 16 | 27.73 ± 0.58% | 26.05 ± 0.71% | 26.27 ± 0.48% | **23.47%** | 25.52 ± 0.04% | 25.74 ± 0.73% |

results dots from the same method. In practice, taking into account the memory burden, the balanced partitioning method of the overall network no longer aims to include the same number of layers in each module; instead, it anticipates guaranteeing each isolated module utilizes as much of the same memory as possible.

The results illustrate that ContSup has a higher memory-performance ratio, meaning that it can accomplish lower error rates with less memory overhead and requires less memory to meet the same precision requirements. Comparing the memory requirements of ContSup and DGL, it can be determined that the memory required to construct the context module (encoder) is very small, but the effect it produces is remarkable; at the same time, the construction of the encoder is not overly complicated, and the network structure can be added directly to the backbone network. Therefore, we believe that ContSup[E] has good scalability and application value, as well as the potential to be further combined and utilized in asynchronous training schemes (Chen et al., 2016; Belilovsky et al., 2020).

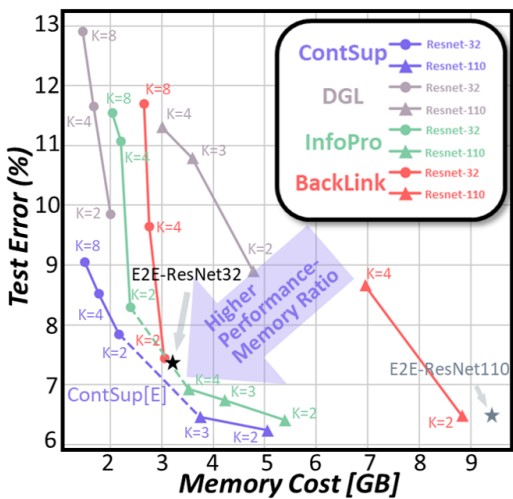

Figure 4: Comparison of the GLL methods' test errors on the CIFAR-10 as a function of GPU memory footprint. Results of training both ResNet32 and ResNet110 on a single Tesla V100-PCIE-32GB GPU are reported.

## 4.3 ABLATION STUDY

**Context Selection.** As shown in Figure 5a, by eliminating the selection of Context, it is evident that the E path plays a crucial role in the R1E structure, which is also consistent with our intuitive assumption (Equation 12). Moreover, the improvement effect of R1 relative to the baseline demonstrates that this local shortcut structure is also useful for addressing the problem of mutual information loss and can be combined with the E method to improve performance as expected.

**Local Decoder for Reconstruction.** As discussed previously in section 2.3, the local reconstruction alleviates the short-sight issue. We conducted a controlled experiment to determine whether a local decoder is compatible with ContSup (Figure 5b). Experiments show that a local decoder can aid in improving the overall performance of ContSup[R1], whereas modes E and R1E are only marginally superior.

**Local Classifier Objective for lower bound of $I(h, y)$.** InfoPro (Wang et al., 2021) first introduced supervised contrastive loss for GLL from the viewpoint of contrastive representation learning as a replacement for cross-entropy loss (Chen et al., 2020; He et al., 2020) and empirically demonstrated its effectiveness over a large batch size. The essence of the two losses for optimizing the lower bound of $I(h, y)$ is identical (see Appendix B for details). Experiments (Figure 5c) indicate that contrastive loss is superior to the cross-entropy loss for protecting $I(h, y)$ from short-sight with large batch size, which is advantageous for the performance optimization of the GLL scheme.

Figure 5: Ablation studies. Test errors of ResNet-32 on CIFAR-10 are reported.

Table 4: Error rate (%) and memory-cost (GB) tradeoffs of training ResNet-32 on CIFAR-10.

| | R0 | | E | | R1 | | R2 | | R4 | | R8 | | R16 | |
|---|---|---|---|---|---|---|---|---|---|---|---|---|---|---|
| $K$ | Error | Mem | Error | Mem | Error | Mem | Error | Mem | Error | Mem | Error | Mem | Error | Mem |
| 8 | 15.70 | 1.54 | 9.50 | 1.76 | 12.77 | 1.88 | 10.91 | 2.43 | 9.80 | 4.25 | 10.56 | 5.25 | - | - |
| 16 | 15.40 | 1.25 | 10.01 | 1.65 | 15.30 | 1.72 | 13.91 | 2.06 | 12.58 | 2.88 | 10.32 | 4.90 | 10.35 | 6.02 |

## 4.4 EXTENSION TOWARD TOPOLOGICAL CONNECTION

The potential expansion of the ContSup structure is shown in Table 4, the greater the density of the Context's connections, the more obvious the accuracy enhancement, but also the greater the computational burden and memory cost. The maximum R$n$ situation of each split case represents the maximum topological connection of $h_l^c$ conveyed by the context, i.e., all $\{h_l^c\}_{l \leq L}$ in the network are connected in pairs in a single forward direction, where ContSup reaches its theoretical limit.

## 4.5 WEIGHT VISUALIZATION

According to the structural design, the manner in which the context selects the historical feature is essentially a shortcut connection; consequently, ContSup's structure can superficially resemble ResNet (He et al., 2016), DenseNet(Huang et al., 2016), and DSnet(Zhang et al., 2021). The implications of cross-layer connections in ContSup may direct us to determine more about how learning occurred locally and provide more explicit, straightforward insights. Several intriguing events are shown in Figure 6.

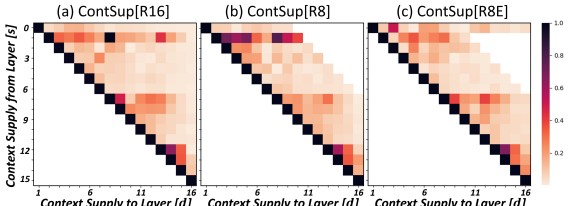

Figure 6: Weight Visualization Study. The average filter weights of $\mathcal{M}$ in trained 16-partitioned ResNet-32 on CIFAR-10. The color of pixel $(s, d)$ encodes the average L1 norm (normalized by the channel number of features) of the weights connecting from layer $s$ to $d$ within ContSup modules $\mathcal{M}$.

Horizontally, we observe that certain features are reused multiple times and that their impact on the deep layer is relatively large (e.g. $s = 1$); this suggests that crucial features can be broadcast into deep layers as parts of ContSup. Vertically, features tend to rely on their neighbors, consistent with the intuitive belief that feature extraction is an iteratively progressive process. Nonetheless, some features are heavily influenced by shallow layers, revealing from the side, that some modules selectively ignore the information transmitted by the most recently connected module and backspace to previous states.

## 5 CONCLUSION

This work indicates and analyzes, from the standpoint of information theory, the bottleneck issue of Greedy Local Learning (GLL) that results in performance degradation. Concluding that existing GLL schemes are incapable of effectively addressing the confirmed habit dilemma, we proposed the Context Supply (ContSup) scheme that enables local modules to retain more information via additional context, thereby enhancing the theoretical effectiveness of final performance. Experiments have demonstrated that ContSup can substantially reduce GPUs memory footprint while maintaining the same level of performance; moreover, relatively stable performance can be maintained even as the number of isolated modules grows, allowing the network to be divided into more segments to reduce memory costs and possibly decomposing module-wise GLL towards layer-wise. ContSup may provide novel opportunities for local learning to reconcile global end-to-end back-propagation with locally plausible biological algorithms.

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

## A    MORE ABOUT MUTUAL INFORMATION IN GLL

Figure 7: Mutual Information Flow in Greedy Local Learning. $I(h_l, r)$ and $I(h_l, y)$ are orthogonal subsets of the information of intermediate feature $h_l$. In practice, the classification loss is defined to maximize $I(\hat{y}_l, y)$, which is short-sighted and may unexpectedly result in a reduction of $I(h_l, y)$. The reconstruction loss is equivalently defined to maximize the entropy $H(h_l)$, which simultaneously maximizes both useful $I(h_l, y)$ and redundant $I(h_l, r)$.

Given intermediate feature $h_l$ outputted by $l^{th}$ module corresponding to the input data $x$ and the label $y$, the local auxiliary module projects feature $h_l$ into target-prediction $\hat{y}_l$ (all of them are treated as random variables). We use $I(A, B)$ as the representation of the mutual information between distribution $A$ and $B$, where $I(h_l, y)$ is naturally designed to measure the amount of task-relevant information in $h_l$, and $I(\hat{y}_l, y)$ represents the predictive correction of the local objective function. Meanwhile, we model the task-irrelevant information in the input data $x$ by introducing the concept of nuisance. A nuisance is defined as an arbitrary random variable that affects $x$ but provides no helpful information for the task of interest Achille & Soatto (2018). Mathematically, given a nuisance $r$, we have $I(r, x) > 0$ and $I(r, y) = 0$, where $y$ is the label. For conciseness, we define $r$ as a maximal nuisance, that $r = argmax_{r^*, I(r^*, y)=0} I(r^*, x)$, then the supposed Markov chain $(y, r) \to x \to ... \to h_l \to ... \to h_L$ is used to describe the forward informational process of stacked modules and information of intermediate feature can be divided into orthogonal factors $H(h_l) = I(h_l, y) + I(h_l, r)$ (see Figure 7).

## B    PROOFS AND EXTRA DISCUSSION ON THEORETICAL RESULTS

In this section, we provide all theorem proofs along with relevant discussion.

### B.1    PROOFS FOR THEOREM 1 IN SECTION 2.2

To prove Theorem 1 of the paper, we first need to introduce a lemma for mutual information in the Markov chain. **Lemma B1 (Data Processing Inequality).** *Suppose that the Markov chain* $X \to Y \to Z$ *holds. Then the mutual information of $X$ and $Z$ is bounded by:*

$$I(X, Y) \geq I(X, Z) \tag{17}$$

**Proof.** If the Markov chain $X \to Y \to Z$ holds, then:

$$p(x, z|y) = \frac{p(x, y, z)}{p(y)} = \frac{p(x, y)p(z|y)}{p(y)} = p(x|y)\,p(z|y) \tag{18}$$

which implies that $X$ and $Z$ are conditionally independent when $Y$ is given, thus:

$$I(X, Z|Y) = 0 \tag{19}$$

Since the mutual information $I(X, Y \cup Z)$ can be expanded as:

$$I(X, Y \cup Z) = I(X, Z) + I(X, Y|Z) = I(X, Y) + I(X, Z|Y) \tag{20}$$

Combining Equation 19 and Equation 20:

$$I(X, Z) + I(X, Y|Z) = I(X, Y) \tag{21}$$

Due to $I(X, Y|Z) \geq 0$, we have:

$$I(X, Y) \geq I(X, Z) \tag{22}$$

The equivalence happened if and only if $I(X, Y|Z) = 0$, which implies that $X \to Z \to Y$ also holds, so that $X \to Y \leftrightarrow Z$ at that time.

Following Lemma B1, two propositions can be derived based on the assumption of Markov chain $(y, r) \to x \to h$ in Greedy Local Learning:

**Proposition B2 (Decreasing potency).** *Suppose that the Markov chain $(y, r) \to x \to ... \to h_{l-1} \to h_l$ holds. Then we have a decreasing upbound of task-related information*

$$I(x, y) \geq I(h_{l-1}, y) \geq I(h_l, y) \tag{23}$$

**Proposition B3 (Local short-sight).** *Suppose that the Markov chain $(y, r) \to x \to ... \to h_l \to \hat{y}_l$ holds. Then the task-related upbound of local prediction is given by:*

$$I(x, y) \geq I(h_l, y) \geq I(\hat{y}_l, y) \tag{24}$$

Note that the serial inference of isolated modules gradually reduces the information entropy, and the amount of task-related information it contains will inevitably diminish (Proposition B2). Moreover, since the local objective of auxiliary modules is defined in terms of the prediction target $\hat{y}_l$, it cannot act directly on the mutual information $I(h_l, y)$ of the intermediate features, resulting in the loss of task-related information (Proposition B3). Those two propositions then constitute Theorem 1 in the main text.

### B.2 Proofs for Theorem 2 in section 2.2

Theorem 2 in the main text is founded on a condition similar to that described in Belilovsky et al. (2019). The mild condition is restated next, followed by a description of the proofs.

**Condition B4 (Potential Identity Mapping).** *Given an arbitrary intermediate feature $h_{l-1}$ and forward procedure $\mathcal{F}^l$, it is always possible to find the parameters $\theta^*$ such that $h_l = \mathcal{F}^l_{\theta^*}(h_{l-1}) = h_{l-1}$. Note that the mild condition can be inherently guaranteed by the residual structure (without downsampling), making it available for partitioned ResNet in GLL.*

**Theorem 2 (Progressive improvement).** *Assume that $\mathcal{F}^l_{\theta_l}$ can be potentially equivalent as an identity mapping as $\mathcal{F}^l_{\theta_l}(h_{l-1}) = h_{l-1}$, then there exists $\hat{\theta}_l$ such that:*

$$\hat{\mathcal{L}}_l\left(\hat{y}_l, y; \theta_l, w_l\right) \leq \hat{\mathcal{L}}_l\left(\hat{y}_l, y; \hat{\theta}_l, w_{l-1}\right) = \hat{\mathcal{L}}_{l-1}\left(\hat{y}_{l-1}, y; \theta_{l-1}, w_{l-1}\right) \tag{25}$$

*Once the local loss $\hat{\mathcal{L}}_l$ is defined on task-relevant mutual information, we obtain:*

$$I(\hat{y}_{l-1}, y) \leq I(\hat{y}_l, y) \tag{26}$$

**Proof.** Assuming Condition B4 holds, given modules $\mathcal{A}_{w_l}$ and $\mathcal{F}^l_{\theta_l}$, there exists $\hat{\theta}_l$ such that:

$$\hat{y}_{l-1} = \mathcal{A}_{w_{l-1}}(h_{l-1}) = \mathcal{A}_{w_{l-1}}(\mathcal{F}^l_{\hat{\theta}_l}(h_{l-1})) = \mathcal{A}_{w_{l-1}}(h_l) = \hat{y}_l \tag{27}$$

therefore,

$$\hat{\mathcal{L}}_l\left(\hat{y}_l, y; \hat{\theta}_l, w_{l-1}\right) = \hat{\mathcal{L}}_{l-1}\left(\hat{y}_{l-1}, y; \theta_{l-1}, w_{l-1}\right) \tag{28}$$

Then, by following the descent direction based on the gradient $\nabla_{\theta_l, w_l}\hat{\mathcal{L}}_l(\hat{y}_l, y)$, a pair of better parameters $(\theta_l, w_l)$ can be found, such that:

$$\hat{\mathcal{L}}_l(\hat{y}_l, y; \theta_l, w_l) \leq \hat{\mathcal{L}}_l\left(\hat{y}_l, y; \hat{\theta}_l, w_{l-1}\right) \tag{29}$$

Thus, there is always a progressive improvement available under Condition B4:

$$\hat{\mathcal{L}}_l(\hat{y}_l, y; \theta_l, w_l) \leq \hat{\mathcal{L}}_l\left(\hat{y}_l, y; \hat{\theta}_l, w_{l-1}\right) = \hat{\mathcal{L}}_{l-1}\left(\hat{y}_{l-1}, y; \theta_{l-1}, w_{l-1}\right) \tag{30}$$

Given the definition of mutual information $I(\hat{y}_l, y)$:

$$I(\hat{y}_l, y) = H(y) - H(y|\hat{y}_l) = H(y) - \mathbb{E}_{(\hat{y}_l, y)}[-\log p(y|\hat{y}_l)] \tag{31}$$

Once the local objective is defined highly correlated with $\mathbb{E}_{(\hat{y}, y)}[-\log p(y|\hat{y}_l)]$, say:

$$\hat{L}(\hat{y}_l, y) \propto -I(\hat{y}_l, y) \tag{32}$$

With Equation 30, we obtain:

$$I(\hat{y}_l, y) \geq I(\hat{y}_{l-1}, y) \tag{33}$$

Finally, Theorem 2 is given by combining Equation 30 and Equation 33.

### B.3 Proofs for Lemma 4 in section 2.3

This section provides full proof for Lemma 4 in the main text.

**Lemma 4 (Entropy Bound).** *Since the Markov chain $x \to ... \to h_{l-1} \to h_l$ is the deterministic procedure in the network forward, the input-related information is bounded by:*

$$H(x) \geq H(h_{l-1}) = I(h_{l-1}, x) \geq H(h_l) = I(h_l, x) = I(h_l, h_{l-1}) \tag{34}$$

*Proof.* Given by the definition of mutual information, we have

$$I(h_l, h_{l-1}) = H(h_l) - H(h_l|h_{l-1}) \tag{35}$$

and,

$$I(h_l, x) = H(h_l) - H(h_l|x) \tag{36}$$

When considering $h_l$ as a deterministic function regards to $h_{l-1}$ (or, to $x$), we obtain $H(h_l|h_{l-1}) = 0$ ($H(h_l|x) = 0$ in a same way), and therefore

$$H(h_l) = I(h_l, h_{l-1}) = I(h_l, x) \tag{37}$$

Following Lemma B1, given the Markov chain $x \to ... \to h_{l-1} \to h_l$, we have:

$$H(x) \geq I(h_{l-1}, x) \geq I(h_l, x) \tag{38}$$

Combining Equation 37 and Equation 38, we obtain:

$$H(x) \geq H(h_{l-1}) = I(h_{l-1}, x) \geq H(h_l) = I(h_l, x) = I(h_l, h_{l-1}) \tag{39}$$

for which we have proven Lemma 4.

### B.4 Further discussion on reconstruction methods in GLL.

Suppose that the Markov chain $(y, r) \to x \to ... \to h_{l-1} \to h_l$ holds, $H(h_l) = I(h_l, r) + I(h_l, y)$ is given by the definition Achille & Soatto (2018). As shown in Figure 8, the objective of maximizing $I(h_l, x) = H(h_l)$ can be precisely divided into two orthometric directions: $I(h_r, r)$ and $I(h_l, y)$. Meanwhile, the local-objective of $I(\hat{y}_l, y)$ is learning to decrease $I(h_l, r)$, which may unexpectedly harm $I(h_l, y)$ (Proposition B3). The short-sight problem caused by local-objective $I(\hat{y}_l, y)$ can be formed into the case when $I(h_l, y)$ decreases, which can be corrected by meticulously accompanying a reconstruction limit (Figure 8).

Despite the fact that local reconstruction can help the local objective recover $I(h_l, y)$, it is still constrained by the upper bound $I(h_l, y) \leq I(h_{l-1}, y)$ (Proposition B2), and trapped in confirmed habits concluded in Theorem 3, where each stacking of greedy modules results in permanent loss of target information. In other words, when the number of isolated partitions is high, the deep modules are unable to recover the information lost in the shallow layers, and the progressive improvement of continuing to stack is limited, resulting in a decline in overall performance.

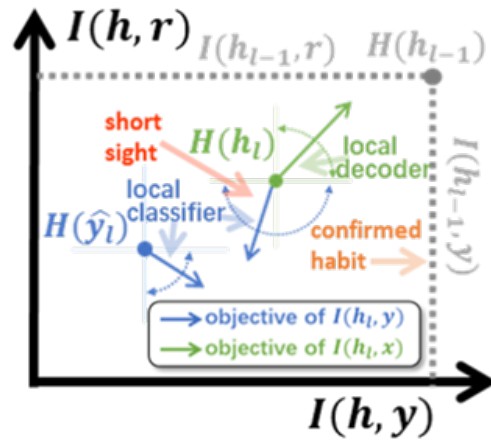

Figure 8: The Learning Directions of Local Objectives. The local classifier is short-sighted since it only considers the function defined on $I(\hat{y}_l, y)$ and may occasionally reduce the expected goal $I(y_l, l)$. The reconstruction objective is intended to maximize both $I(h_l, r)$ and $I(h_l, y)$ and to neutralize short-sighted local optimization.

## B.5 FURTHER DISCUSSIONS ON LOCAL CLASSIFIER OBJECTIVES

Local objectives for $I(h, y)$ are required by GLL, and they are always realized via the creation of auxiliary classifiers that evaluate task-relevant information. Define the local classifiers for maximum likelihood estimation of $p(y|h)$, such that $q(\hat{y}) = \hat{p}_{\mathcal{A}_w}(y|h) \approx p(y|h)$. The GLL methods often choose the widely used cross-entropy loss for local classifiers Belilovsky et al. (2019); Zhang et al. (2022); Duan & Principe (2022); Guo et al. (2023), which can be expressed as:

$$\mathbb{E}\big[\hat{L}_{CE}(\hat{y}, y)\big] = \mathbb{E}_h\big[-y \log q(\hat{y})\big] = \frac{1}{N}\big[\sum_{}^{N} -y \log q(\hat{y})\big] \tag{40}$$

Then, we could demonstrate that minimizing the cross-entropy loss is exactly maximizing the lower bound of task-relevant information $I(h, y)$. Note that

$$H(y) - I(h, y) = H(y|h) \tag{41}$$

$$= \mathbb{E}_{(h,y)}\big[-\log p(y|h)\big] \tag{42}$$

$$= \mathbb{E}_h\Big[\mathbb{E}_y\big[-p(y|h)\log p(y|h)\big]\Big] \tag{43}$$

$$\leq \mathbb{E}_h\Big[\mathbb{E}_y\big[-p(y|h)\log q(\hat{y})\big]\Big] \quad \text{(Gibb's inequality)} \tag{44}$$

$$= \mathbb{E}_h\Big[\frac{1}{C}\big[\sum_{c=1}^{C} -\mathbb{1}_{y_c=1}\log q(\hat{y})_c\big]\Big] \tag{45}$$

$$= \frac{1}{C}\Big[\mathbb{E}_h\big[-y\log q(\hat{y})\big]\Big] \tag{46}$$

$$= \frac{1}{C}\mathbb{E}\big[\hat{L}_{CE}(\hat{y}, y)\big] \tag{47}$$

In the above, Equation 45 samples the probabilities along class dimension with the number of classes $C$, where the value exists along on the target class. Finally, we have $I(h, y) \geq H(y) - \frac{1}{C}\mathbb{E}\big[\hat{L}_{CE}(\hat{y}, y)\big]$, and thus minimizing $\hat{L}_{CE}(\hat{y}, y)$ under the stochastic gradient descent framework maximizes a lower bound of $I(h, y)$.

Meanwhile, as mentioned in section 4, the supervised contrastive loss, namely $L_{contrast}$, is first applied to GLL by Wang et al. (2021) from the viewpoint of contrastive representation learning as a replacement for cross-entropy loss Chen et al. (2020); He et al. (2020). Contrastive loss is founded on supervised clustering utilizing latent features; therefore, its loss is defined by the relative distance between any two samples within the same batch, as

$$L_{contrast} = \frac{1}{\sum_{i \neq j} \mathbb{1}_{y^i=y^j}} \sum_{i \neq j} \Big[\mathbb{1}_{y^i=y^j} \log \frac{exp((\hat{y}^i)^T \hat{y}^j / \tau)}{\sum_{k=1}^{N} \mathbb{1}_{i \neq k} exp((\hat{y}^i)^T \hat{y}^k / \tau)}\Big], \quad \hat{y}^b = \mathcal{A}_w(h^b) \tag{48}$$

Table 5: Architecture of the local decoder $\mathcal{W}$ on CIFAR-10, SVHN and STL-10.

| Input: 32×32 / 16×16 / 8×8 feature maps (96×96 / 48×48 / 24×24 on STL-10) |
| :---: |
| Bilinear Interpolation to 32×32 (96×96 on STL-10) |
| 3×3 conv, stride=1, padding=1, output channel=12, BatchNorm + ReLU |
| 3×3 conv, stride=1, padding=1, output channel=3, Sigmoid |

Table 6: Architecture of the local classifier $\mathcal{A}$ on CIFAR-10, SVHN and STL-10.

| Input: 32×32 / 16×16 / 8×8 feature maps (96×96 / 48×48 / 24×24 on STL-10) |
| :---: |
| 32×32 (96×96) input features: 3×3 conv, stride=2, padding=1, output channel=32, BatchNorm + ReLU
16×16 (48×48) input features: 3×3 conv, stride=2, padding=1, output channel=64, BatchNorm + ReLU
8×8 (24×24) input features: 3×3 conv, stride=1, padding=1, output channel=64, BatchNorm + ReLU |
| Global average pooling |
| Fully connected 32 / 64 → 128, ReLU |
| Fully connected 128 → 10 for $L_{CE}$ or 128 → 128 for $L_{contrast}$ |

The effectiveness of contrastive loss in GLL over a large batch size is empirically demonstrated. It was also proofed that $\mathbb{E}\big[L_{contrast}\big] \geq \log(N-1) - I(h, y)$ Wang et al. (2021); consequently, contrastive loss also maximizes the lower bound of $I(h, y)$ and offers an alternative to cross-entropy in practice.

## C  EXPERIMENTAL DETAILS

### C.1  BENCHMARK DATASETS

In this paper, three image datasets (CIFAR-10 (Krizhevsky, 2009), SVHN (Netzer et al., 2011), and STL-10 (Coates et al., 2011)) are used in the experiments to evaluate the efficacy of ContSup. Here, we describe the composition of each dataset and the pre-processing methods applied to them. (1) CIFAR-10 (Krizhevsky, 2009) consists of 60,000 32x32 colored images of 10 classes, with 50,000 for training and 10,000 for testing. According to the standard pre-processing method, we normalize the origin images with channel means and standard deviations, followed by 4x4 random translation and random horizontal flip (He et al., 2016; Huang et al., 2016). (2) SVHN (Netzer et al., 2011) consists of 32x32 real-world images of house numbers, 10 classes for digits ranging from 0 to 9, with 73,257 images for training and 26,032 images for evaluation. Random 2x2 translation is performed to augment the training set, as (Tarvainen & Valpola, 2017). (3) STL-10 (Coates et al., 2011) contains 96×96 labeled RGB images belonging to 10 classes, which are split into 5,000 training samples and 8,000 test samples. We train with all of the labeled images and evaluate performance on the test set. Data augmentation is performed by a 4x4 random translation followed by a random horizontal flip.

### C.2  TRAINING HYPER-PARAMETERS

The experiments use the ResNet-32 and ResNet-110 networks (both of which are part of the ResNet architectural family proposed in (He et al., 2016)) as their basis. We employ an SGD optimizer with a Nesterov momentum of 0.9 and an L2 weight decay ratio of 1e-4 to train the networks for 160 epochs. The initial learning rate is 0.8 for CIFAR-10/SVHN and 0.1 for STL-10, while the batch size is 1024 for CIFAR-10/SVHN and 128 for STL-10. It is important to note that the experimental setups described here are employed by all of the GLL methods reported in Table 2 and Table 3 in Section 4.2.

Table 7: Architecture of the input-based context supply $\mathcal{M}_E$ on CIFAR-10, SVHN and STL-10.

| Input: 32×32 origin input (96×96 on STL-10) |
| --- |
| 3×3 conv, stride=1, padding=1, output channel=16/32/64, BatchNorm + ReLU |
| 3×3 conv, stride=1, padding=1, output channel=16/32/64, BatchNorm + ReLU |
| Adaptive average pooling |
| Output: 32×32 / 16×16 / 8×8 feature maps (96×96 / 48×48 / 24×24 on STL-10) |

Table 8: Architecture of the feature-based context supply $\mathcal{M}_R$ on CIFAR-10, SVHN and STL-10.

| Input: 32×32 / 16×16 / 8×8 feature maps (96×96 / 48×48 / 24×24 on STL-10) |
| --- |
| 1×1 conv, stride=1, padding=0, output channel=16/32/64, BatchNorm + ReLU |
| Adaptive average pooling |
| Output: 32×32 / 16×16 / 8×8 feature maps (96×96 / 48×48 / 24×24 on STL-10) |

### C.3 IMPLEMENTATIONS OF AUXILIARY MODULES

In this section, we describe the experimental implementations of $\mathcal{F}_\theta$, $\mathcal{A}_w$ and $\mathcal{M}_\phi$ that mentioned in the main text, as well as an additional decoder $\mathcal{W}$ for reconstruction objective $I(h, x)$. The structures of applied modules are identical to that of other GLL methods (see Table 5 and Table 6) (Belilovsky et al., 2020; Wang et al., 2021; Guo et al., 2023); while the architecture of proposed module $\mathcal{M}$ for context supply is shown in Table 7 and Table 8. Note that $\mathcal{F}$ is obtained by dividing Resnet into gradient-isolated modules, which is discussed further with the partitioning strategy. Decoder $\mathcal{W}$ reconstructs the original images from intermediate features. The output size of auxiliary classifier $\mathcal{A}$ varies depending on whether the cross-entropy loss $L_{CE}$ or the contrastive loss $L_{contrast}$ is applied. Two distinct forms of $\mathcal{M}$ are used for $\mathcal{M}_E(x)$ and $\mathcal{M}_R(h_{past})$, respectively.

### C.4 PARTITIONING STRATEGY ON RESNET

The GLL method necessitates partitioning the entire network into K gradient-isolated modules, and the partitioning strategy must be clarified. For simplicity, we are only considering about ResNets that exist in our experiments. Due to the impossibility of subdividing the residual blocks into smaller parts, we regard each residual block inside ResNets to be an indivisible minimal unit. Particularly, the entire network's first convolutional layer is considered an additional minimal unit, comparable to residual blocks. In this instance, ResNet-32 has a total of 15 potential partitioning locations, so $K = 16$ is sufficient to partition all minimal units into isolated modules. Then, it is straightforward for cases of $K = 2/4/8$, which only need to ensure that each isolated module contains the same number of minimal units. For ResNet-110 with 55 minimal units, one less minimal unit is assigned to earlier isolated modules when the number of minimal units is not evenly divisible by $K$, such that $\{27, 28\}$ for $K = 2$, $\{13, 14, 14, 14\}$ for $K = 4$, etc. Notably, when we concentrate on reducing memory costs, we employ a distinct partitioning strategy, namely the memory balance strategy; in this case, the networks are meticulously partitioned such that each local module incurs comparable memory costs during training, as described in Section 4.2.

## D VISUALIZATION OF TASK-RELEVANT INFORMATION INSIDE INTERMEDIATE FEATURES

In order to provide evidence to support the theoretical analysis, we have included the estimate and visualization of task-relevant information pertaining to intermediate features, as shown in Figure 9. The experiment was performed using an 8-partitioned Resnet32 on CIFAR-10. In this case, each block consists of two units, with each unit representing the smallest indivisible unit as spec-

ified in Appendix C.4. The hyperparameter remains consistent. The models exhibiting superior performance during the training phase are saved, and then utilized to estimate the feature. In order to estimate task-related information, we adopt the method employed in InfoPro. Specifically, we utilize the estimation of $I(h, y)$ by leveraging the equation $I(h, y) = H(y) - H(y|h) = H(y) - E_{(h,y)}[-\log p(y|h)]$. To achieve this, we train a supplementary classifier $q_\phi(y|h)$ with parameters $\phi$ to serve as an approximation of $p(y|h)$. Consequently, we can express $I(h, y)$ as an approximation given by $I(h, y) \approx max_\phi H(y) - \frac{1}{N}[\sum_{i=1}^{N} - \log q_\phi(y_i|h_i)]$. This estimation has a strong correlation with the value of $-\frac{1}{N}[\sum_{i=1}^{N} - \log q_\phi(y_i|h_i)]$, which can be interpreted as the maximum level of performance achieved by a classifier that is based on h. In our setting, ResNet-32 is used as the $q_\phi$, including upsampling and channel-alignment in the first layer.

Among the three pure schemes, it is evident that the mutual information of ContSup exhibits significant benefits in the deep layer. The pivotal moment occurs when contextual information is provided subsequent to minimum units # 7, hence increasing the disparity with InfoPro. To address the issue of bottleneck caused by confirmed habits, we devised and implemented hybrid schemes with training the first 2 or 4 modules using coarse DGL and achieved the phenomena of decreasing potency, as stated in Theorem 1.1 in the main text. In this case, while InfoPro can only mitigate the phenomenon of information decline to the greatest extent feasible, ContSup effectively addresses the challenges posed by the confirmed habits, as expected.

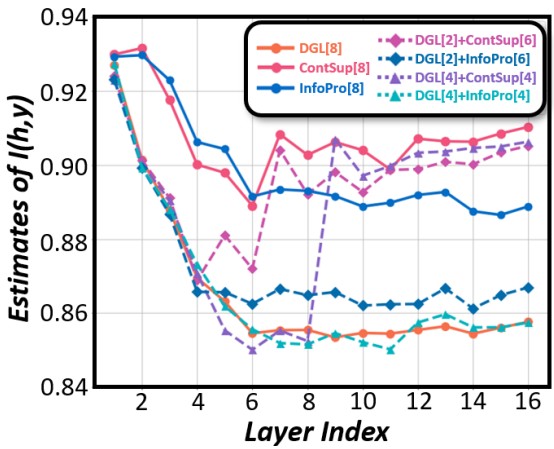

Figure 9: Visualization results of estimating task-relevant information through intermediate features of 8-partitioned ResNet-32 on CIFAR-10. "ContSup[8]", "InfoPro[8]", and "DGL[8]" denote the pure implementations of each GLL method. The notations in the form of "A[x]+B[y]" represent hybrid methods that employ method A for the first x blocks and method B for the remaining y blocks.

# E MORE DISCUSSION
## ABOUT EXPERIMENTAL RESULTS

In section 4.2, we compare ContSup's performance to that of DGL, InfoPro, and BackLink on a variety of image classification benchmarks. The experimental results show that, when the number of network segmentations increases, the performance of several different approaches shows a clear negative trend, indicating that the number of segmentations poses an essential obstacle to the further expansion of GLL methods. Among comparable GLL methods, Backlink allows overlap between separated modules, making it superior to the non-overlapping scheme within the same number of segmentations; its benefit is evident when the number of segments is small, but it remains vulnerable to the confirmed habit dilemma as $K$ rises. InfoPro improves upon baseline performance by introducing an additional local decoder to boost information entropy; however, it incurs additional computational and memory costs that partially offset the GLL's benefits. In contrast, the proposed ContSup effectively mitigates the decreasing trend in a large number of segmentations, by introducing a simple route for context supply. Furthermore, we also compare the GPU memory cost of ContSup to those of InfoPro, BackLink, DGL, and E2E. It can be determined that the memory required to construct the context module (encoder) is very small, but the effect it produces is remarkable; at the same time, the construction of the encoder is not overly complicated, and the network structure can be added directly to the backbone network. Intriguingly, our encoder structure from $x$ to local feature is the inverse process of the designed decoder structure in InfoPro, which expects local features to reconstruct $x$ for larger $I(h, x)$; however, our encoder structure is lean, whereas the decoder structure introduces about much memory overhead for reconstruction. Therefore, we believe that the encoder can take into consideration the prospective alternative of the decoder while avoiding the time-consuming step of modifying hyperparameters to balance classifier and decoder objectives. In section 4.4, we extend the context mode from R1 to Rn, such that $[h_{l-n}^c, ..., h_{l-1}^c]$ are used as context to supplement the existing mutual information of $h_l$. Obviously, this linear superposition scheme is inefficient, but it does provide a reasonable comprehension of the ContSup structure. The experimental results are intuitive, i.e., as

Table 10: Evaluation of inference overhead. The inference costs over the whole test datasets (in seconds) are evaluated on CIFAR-10 with $K$-partitioned ResNet-32. The trial is built on a single NVIDIA GeForce RTX 4090 GPU.

| Backbone | K=2 | | K=4 | | K=8 | | K=16 | |
|---|---|---|---|---|---|---|---|---|
| | E | R1E | E | R1E | E | R1E | E | R1E |
| Resnet32 | 1.301 / | 1.305 0.31% | 1.307 0.46% | 1.321 1.54% | 1.337 2.77% | 1.367 5.07% | 1.414 8.69% | 1.437 10.45% | 1.513 16.30% |
| Resnet110 | 1.611 / | 1.614 0.19% | 1.619 0.50% | 1.628 1.06% | 1.632 1.30% | 1.657 2.86% | 1.683 4.47% | 1.735 7.70% | 1.805 12.04% |

$n$ increases, accuracy continues to rise and tends to plateau. One may discover that the ContSup[E] structure appears to be superior to R$n$; this is not surprising given that the encoder's design structure is more complex, consisting of two $3 \times 3$ convolution operations, whereas R$n$'s design structure is much simpler, consisting of a $1 \times 1$ convolution (linear combination of channels) for dimension alignment.

## F  MORE RESULTS

**Combination with other GLL variations.** We are well aware that combining ContSup and the existing approaches can help to demonstrate the scalability of ContSup while compensating for the performance in small K. ContSup is developed specifically for the GLL framework, therefore making it potentially suitable to other GLL variations. Since ContSup is designed for the GLL framework, it is theoretically applicable to multiple GLL variants. As in experimental section, some GLL designs that improved by InfoPro and DGL have been directly taken into consideration in the proposed ContSup scheme. Figure 5(b,c) illustrates the effect of those optimizations employed on ContSup, demonstrating that these optimizations can be maintained in the implementation of ContSup. In light of the scheme's scalability, we conducted a trial wih error rates on CIFAR-10, as shown in Table 9. It can be found that the BackLink and ContSup exhibits compatibility with one another. The incorporation of BackLink into the ContSup scheme has the potential to significantly enhance the performance of small K. In accordance with the empirical results shown in Table 3, ContSup does not weaken the original performance when K is small; however, the influence of the confirmed habits issue is less pronounced in this scenario, and the performance gains achieved by ContSup are constrained. Hence, it is vital to thoroughly contemplate additional bottleneck elements that restrict GLL in this case. Incorporating existing approaches might serve as a first step.

Table 9: Test errors of GLL combination. The experiment is evaluated on CIFAR-10 with $K$-partitioned ResNet-32.

| | K=2 | K=4 |
|---|---|---|
| BackLink | 6.97 | 7.55 |
| ContSup[E] | 8.04 | 8.75 |
| ContSup[E] + BackLink | 6.88 | 7.18 |

**Inference Cost from Context Modules.** The introduced extra parameters of the local encoders to compress the context cannot be discarded in inference time, leading to a reduction in computing efficiency. Fortunately, the size of the context module is comparatively less than that of the backbone network, which helps in reducing the negative effect of a decrease in inference speed. The cost of inference will increase to some degree, for instance, on CIFAR-10, the increasing inference time (in seconds) for Resnet-32 and Resnet-110 is shown in Table 10. Thorough evaluation and optimization at the application level still need careful consideration.

**Potential Improvement of Context Selection.** Based on our theoretical study, the primary function of context is to compensate for the absence of task-related information in order to overcome the challenge of proven habits. However, the issue of optimizing context selection still remains unresolved and requires additional attention. It is posited that the pruning operation has the po-

Table 11: Test errors with different context selection. The experiment is evaluated on CIFAR-10 with 16-partitioned ResNet-32.

| | R0 | E | R1E | M1R1E | M2R1E | R8E |
|---|---|---|---|---|---|---|
| K=16 | 15.30 | 10.21 | 9.98 | 9.71 | 9.22 | 9.42 |

Table 12: Comparison of training overhead under different GLL methods. The batch-wise training times (in seconds) are evaluated on CIFAR-10 with $K$-partitioned ResNet-32. The trial is built on a single NVIDIA GeForce RTX 2070 GPU.

|       | DGL   | InfoPro | ContSup[E] | ContSup[R1E] | BackLink[l=4] |
|-------|-------|---------|------------|--------------|---------------|
| K=2   | 0.727 | 0.778   | 0.737      | 0.740        | **0.817**[l=4] |
| K=4   | 0.738 | **0.917** | 0.795    | 0.819        | 0.910[l=4]    |
| K=8   | 0.771 | **1.235** | 0.901    | 0.959        | 1.023[l=2]    |
| K=16  | 0.829 | **1.599** | 1.178    | 1.227        | 1.099[l=2]    |

tential to be executed based on the comprehensive topological connectivity of RnE. Given the comparative simplicity involved in processing the next module's feature (R), it is conceivable that its approximation might make it redundant. On the other hand, the origin input (E) is more rudimentary and hard to process, although it offers a higher quantity of information. Consequently, incorporating the intermediate features between these two could potentially be advantageous. Based on the aforementioned assumption, for module $l$, it is possible to choose the intermediate point as the context by considering R1E, which includes the $0^{th}$ origin input and $(\frac{(l-2)}{2})^{th}$ additional features. Examples of suitable contexts include the bisector at $\frac{(l-2)}{2}$, , as well as two-three points at $\frac{(l-2)}{3}$ and $\frac{2(l-2)}{3}$. The scheme is denoted as Mi, where i is the number of chosen intermediate points. The experimental results for error rates are shown in Table 11. The context module for the intermediate feature M is designed according to R, which consists of a single 1*1 convolution and pooling. This design decision ensures channel and dimension alignment, as shown in Table 8. It has been observed that both M1R1E and M2R1E exhibit a heightened level of performance that closely resembles that of R1E when K=8. Acknowledging the limited depth, the inclusion of trivial examples serves to illustrate the possibility of enhanced optimization in the process of context selection.

**Comparison of Training Overhead.** For the compared method, InfoPro, the decoder's hyperparameters must be altered and optimized through repetitive attempts in order to effectively safeguard the information entropy, which increases the trial cost. ContSup, on the other hand, does not depend on the decoder's design, and the Context module definition contains no additional hyperparameters; therefore, it is simple and effective for experimental deployment. Due to the local classifier, DGL will increase the training overhead in GPU training compared to E2E. On this basis, the local-decoder structure introduced by InfoPro will further increase the training overhead; instead, the local-encoder design in ContSup is lighter than that of the decoder, so its training speed is even faster than that of InfoPro. For instance, we compare the batch-wise training time (in seconds) of ResNet32 on CIFAR-10, and the results are shown in Table 12 [with baseline E2E=0.721].

## G LIMITATION AND FUTURE WORK

The context selection within the ContSup structure has not been fully developed. Experience has shown that a straightforward implementation of context can significantly boost the performance and memory footprint of the GLL scheme on the GPU. In addition, we investigate the expansion potential of ContSup and its expected upper bound, as described in Section 4.4, which is based on a linear expansion that is memory-intensive and difficult to expand. On the basis of this, an optimized expansion strategy may be considered by tailoring wholly topological connections out of superfluous connections, so that context selection prioritizes only the most important relationships and avoids similar or redundant features. In addition, future research may investigate the relationship between the structural design of the context and the extant networks in order to draw further conclusions. Using the analogy and practice of the GLL training scheme may also help analyze the possible interpretation of the E2E training scheme in an effort to discover valuable insights for enhancing the interpretability of neural networks.

