# OpenReview forum: "Go beyond End-to-End Training: Boosting Greedy Local Learning with Context Supply"
_ICLR.cc/2024/Conference — ICLR 2024 Conference Withdrawn Submission_

### Official Review · Reviewer_qZKx · 2023-10-29

**Soundness:** 3 good
**Presentation:** 2 fair
**Contribution:** 2 fair
**Rating:** 3
**Confidence:** 5

**Summary:**

This paper proposes an improved locally supervised learning method, which is named as Context Supply (ContSup). ContSup is motivated by the phenomenon of short-sight task-relevant information loss, i.e., in the earlier modules, introducing local supervision signals tend to encourage the loss of important task-relevant information, such that the performance of later modules is limited by the insufficiently informative inputs. The authors propose to alleviate this problem by incorporating context supply between isolated modules to compensate for information loss. Experimental results on CIFAR-10/100, SVHN, and STL-10.

**Strengths:**

1. The writing of this paper is good.
2. Although InfoPro has studied the phenomenon of short-sight task-relevant information loss, the authors present a further theoretical analysis of this issue, which I think is an acceptable contribution.
3. The proposed method seems well-motivated and effective.

**Weaknesses:**

1. The experiments are insufficient. Some results on ImageNet or the results of tasks in addition to only image classification will make this paper stronger. Notably, given that ImageNet is one of the most reasonable benchmarks to evaluate deep learning algorithms, the real contributions of this paper are still questionable.
2. According to Table 10, ContSup increases the inference cost significantly. I think this is an important problem. This issue is especially serious when K is large, where the authors claim that ContSup is mainly useful for. The comparisons with almost all the baselines are unfair, since the model used by ContSup is enlarged significantly. In other words, the authors think ContSup is mainly useful when K is large, but the comparisons are extremely unfair in these cases. In contrast, when K is small, the gains of ContSup are very marginal (in fact, here, the comparisons are still unfair).
3. In terms of Table 10, the results of additional FLOPs during inference should also be provided. Both Table 10 and these new results should be highlighted in the main text.
4. I politely disagree with the authors on the training cost (Table 12). Other baselines (e.g., DGL, InfoPro) can also use light auxiliary module designs. ContSup also needs to adjust hyper-parameters (e.g., the architecture of auxiliary nets).

**Questions:**

See weaknesses.

---

### Official Review · Reviewer_DP2z · 2023-10-31

**Soundness:** 3 good
**Presentation:** 2 fair
**Contribution:** 3 good
**Rating:** 6
**Confidence:** 2

**Summary:**

This work focuses on greedy local learning (GLL), which segments a full network into modules and trains different modules locally (gradient only flows within each module). One major challenge of the previous GLL scheme is that when more modules are required for deep networks, the local learning capacity degrades. This paper proposes a scheme called ContSup, which includes context supply between different modules to compensate for the information loss based on the analysis of information theory. Extensive experiments on various datasets demonstrate the effectiveness and efficiency of the proposed scheme.

**Strengths:**

* The motivation of this work is clear. GLL is an important learning scheme and a promising alternative for E2E training. The author theoretically analyzes the challenge of the current GLL scheme and points out the dilemma of confirmed habits. Context supply is then proposed to solve this challenge.

* There are many theoretical analysis based of information theory, which improves the soundness of this work.

* Experiment design is comprehensive, covering various model and dataset.

**Weaknesses:**

* Presentation quality needs to be improved, including but not limited to:
  * Some sections are not clear (In section 2.1 what is the difference between main parameters $\theta_l$ and primary parameters $w_l$?)
  * Some papers are not correctly cited (For example, "Densely Connected Convolutional Networks" CVPR version should be cited instead of arxiv, "Revisiting Locally Supervised Learning: an Alternative to End-to-end Training" ICLR version should be cited instead of arxiv, etc.)
  * Figure 4 is also not clearly presented and it's difficult to compare different GLL methods. (I suggest using a table instead or putting ResNet-32 and ResNet-110 results seperately in two subfigures. Also, K=3 for BackLink on ResNet-110 seems missing?)
  * Figure 8 in the appendix is not clear both visually and conceptually.

* As the author mentioned in Sec 4.2, the ContSup are not effective with a small partition number $K$. Even when $K$ is larger (8, 16), the improvement of test accuracy is also incremental (compared to BackLink).

**Questions:**

Please see the weakness section above.

---

### Official Review · Reviewer_PbnD · 2023-11-01

**Soundness:** 3 good
**Presentation:** 3 good
**Contribution:** 3 good
**Rating:** 5
**Confidence:** 4

**Summary:**

The paper introduces ContSup, a new method for greedy local learning (GLL). ContSup builds on the observation that, with increasing numbers of gradient-isolated modules, vanilla GLL approaches tend to lose task related information. ContSup aims to alleviate the problems by incorporating additional inputs to each gradient-isolated module, ensuring the flow of relevant information from adjacent modules as well as direct model input. Compared to the baselines, ContSup demonstrates superior performance on the CIFAR10, SVHN, and STL-10 benchmarks.

**Strengths:**

(S1) [Motivation] The paper presents a valuable exploration of greedy local learning (GLL), which could potentially provide a memory-efficient alternative to the conventional end-to-end learning paradigm.

(S2) [Analysis] The paper identifies the performance bottleneck of vanilla GLL approaches: the loss of task-specific information as the number of gradient-isolated partitions increases. This observation is supported by theoretical analysis.

(S3) [Ablation] The paper provides extra ablations on context selection, context reconstruction, and local objectives.

**Weaknesses:**

(W1) [Method] The reviewer has concerns about the potential increase in training and inference overheads due to ContSup. i) during training, ContSup relies on contexts from adjacent modules, making it difficult to train the individual modules separately. This seems to violate the original objective of GLL. ii) ContSup introduces new modules to the original architecture which may increase the inference latency.


(W2) [Evaluation] One of the main goals of GLL is to reduce the computing and memory overheads when training large models on large datasets. In the paper, ContSup is evaluated only on smaller benchmarks, i.e.CIFAR-10, SVHN, STL-10, which may not fully represent its efficacy. The authors are encouraged to extend the evaluations to larger benchmarks such as ImageNet1k, to better assess ContSup’s scalability and potential impact. Moreover, in line with (W1), it would be beneficial to include an analysis of the inference latency.


Overall, the reviewer appreciates the authors’ exploration on GLL and the valuable analysis on the performance bottleneck. However, the reviewer has concerns about the extra training and inference overheads introduced by the proposed method, and believe that the paper could benefit from more comprehensive evaluations. At this moment, the reviewer rates the paper as marginally below the acceptance threshold.

**Questions:**

N.A.